# Influence of New Sleeve Composite on Fracture Behavior of Anterior Teeth with Flared Root Canals

**DOI:** 10.3390/polym14194073

**Published:** 2022-09-28

**Authors:** Shinji Yoshii, Sufyan Garoushi, Chiaki Kitamura, Pekka K. Vallittu, Lippo Lassila

**Affiliations:** 1Department of Biomaterials Science and Turku Clinical Biomaterial Center-TCBC, Institute of Dentistry, University of Turku, 20520 Turku, Finland; 2Division of Promoting Learning Design Education, Kyushu Dental University, Fukuoka 803-8580, Japan; 3Division of Endodontics and Restorative Dentistry, Kyushu Dental University, Fukuoka 803-8580, Japan; 4City of Turku Welfare Division, Oral Health Care, 20520 Turku, Finland

**Keywords:** sleeve composite, flared root canals, fiber post, loading test

## Abstract

We evaluated the fracture strength and failure mode of non-ferrule teeth with flared root canals that were restored using new experimental sleeve composites. Fifty endodontically treated anterior teeth with flared root canals were restored with direct restorations utilizing different techniques. Group A had teeth (non-ferrule) restored using commercialized MI glass fiber post + Gradia Core as core build-up. Group B had teeth (non-ferrule) restored with commercialized i-TFC glass fiber post + sleeve system. In Group C, the teeth (non-ferrule) were restored with an experimental sleeve composite with commercialized MI glass fiber post and Gradia Core. Group D, teeth (non-ferrule), were restored using custom-made tapered E-glass filling post and Gradia Core. Group E, teeth (with ferrule), were restored with commercialized MI glass fiber post + Gradia Core. After core construction, all specimens underwent direct composite crown restoration and were loaded until fracture using a universal testing machine. Average fracture loads were compared, and the failure modes were observed. Group C exhibited significantly greater fracture strength than other groups (*p* < 0.05). Favorable fracture teeth ratio of group C was more than that of the other groups. Thus, the new experimental sleeve composite could be clinically useful for core construction of non-ferrule teeth.

## 1. Introduction

The use of glass fiber posts and composite resin for the core construction of endodontically treated teeth has been favored over the previously common metallic casted posts [1,2]. This emerging preference may be attributed to their reported prevention of root fracture as well as their similarity to dentin in terms of mechanical properties and behavior [3,4].

Various construction methods have been developed to fill root canals with fiber posts [5,6,7,8]. For example, several reports have indicated that the risk of root fracture decreases when the root canal is filled with fiber posts, as well as sleeve and accessory fiber posts [9,10,11,12]; furthermore, specific fiber-post arrangements can reinforce the match of the fiber post to the root canal as well as its retention therein [13,14].

Filling the post space with fibers is advantageous in that it enhances adhesion by reducing the amount of resin in the post space, thereby suppressing the polymerization shrinkage of the resin; the reinforcement of the fiber post further contributes to this adhesion [15,16]. However, the risk of vertical root fracture remains a serious clinical problem, despite the successful restoration of the endodontically treated tooth using these fiber-post arrangements [17,18].

One of the factors underlying the increased risk of vertical root fracture is the absence of ferrules in teeth: the presence of the ferrule is thought to extend the tooth life, a supposition supported by the frequency with which non-ferrule teeth are seen in clinical practice [18,19,20].

Although many fiber posts have featured unidirectional fibers, experimental sleeve composites of three-dimensional woven structures have previously demonstrated a promising performance; this novel structure exhibits the same flexural modulus as thin dentin and can be customized easily into various shapes [21]. Furthermore, this sleeve composite showed the highest flexural modulus and maximum bending stress in comparison with commercial fiber post systems. Concurrently, no data are available on the influence of this experimental sleeve composite structure on the fracture behavior of anterior teeth with flared root canals. Therefore, the present study aimed to assess the fracture behavior of non-ferrule anterior teeth with flared root canals restored with sleeve composite compared with commercially available fiber post systems. The null hypotheses were that (1) the anterior teeth restored with the tested restorative techniques would show similar fracture load values, and that (2) the failure mode would not depend on the applied restorative technique.

## 2. Materials and Methods

### 2.1. Materials

Table 1 shows the materials used in this study. We assessed a total of 50 extracted maxillary central teeth that were extracted due to periodontal disease without endodontic treatment preserved in choloramine T at longest 3 months before being used. All specimens were sectioned at 18 mm from the apex with a low-speed diamond saw (Secotom-50, Struers, Tokyo, Japan), and the crowns were removed. According to the X-ray, all teeth had only one root canal. Additionally, working Length were determined at 17 mm. Root canal preparations were performed using K-files up to No. 60 (Maillefer K-file, Dentsply Sirona, York, PE, Canada) and were irrigated with 3% sodium hypochlorite solution (Canasol, Magnum Dental AS, Tartu, Estonia), 8.5% EDTA (EDTA, Magnum Dental AS, Tartu, Estonia), and distilled water. The specimens were subsequently dried, and the root canals were obturated with calcium hydroxide root canal sealer (AH plus, Dentsply Sirona, York, PE, Canada) and obturated material (GUTTA PERCHA POINTS, TOP DENT. Espoo, Finland). After 24 h, the lengths of 40 teeth were shortened to 15 mm, and a post space of 10 mm in depth was prepared using a diamond instrument (TF-13, MANI, Tochigi, Japan) with water coolant. The thickness of the dentin wall was adjusted to 1 mm at the cervical area over the entire circumference. Measurements were performed using digital calipers (Digital caliper, Diesella Co., Marsvej 20, Kolding, Denmark). These specimens were divided into four groups of 10 specimens each (Figure 1): the MI, i-TFC, sleeve composite, and custom-made tapered groups; the remaining 10 teeth were included in the ferrule group. The sample size was determined after referring to the preliminary study. All specimens were evaluated for fissures and fractures after the canals were prepared by agreement of two examiners without microscopy.

### 2.2. Experimental Groups

Group A (MI group): The lengths of the prefabricated glass fiber posts (MI post, GC Europe, Leuven, Belgium) were adjusted to 15 mm. The posts were applied with 40% phosphoric acid gel (Scotchbond, Universal Etchant, 3M ESPE, St. Paul, MN, USA) for 10 s, rinsed with water and air-dried, and then coated with a silane coupling agent (Ceramic primer II, GC Europe, Leuven, Belgium). A bonding agent (G-Premio bond, GC Europe, Leuven, Belgium) was subsequently applied to the dentin in the root canal wall and the cervical shoulder regions at the top of the root for 10 s. The specimens were cured with a light cure unit (Elipar S10, 3M ESPE, St. Paul, MN, USA) at an intensity of 450 mW/cm^2^ for 20 s. Dual cure resin cement (Gradia Core, GC Europe, Leuven, Belgium) was injected to fill the post space into which the post was then inserted and simultaneously built up the core. The curing light was applied to the occlusal, buccal, and lingual sides for 20 s each. The abutment was manually prepared to a height of 5 mm from the cervical region using a diamond bur (EX-17, MANI, Tochigi, Japan) (Figure 1A).

Group B (i-TFC group): the post of this system (i-TFC post, Sun Medical Co., Ltd., Shiga, Japan) was adjusted to 15 mm, whereas the sleeve (i-TFC sleeve, Sun Medical Co., Ltd., Shiga, Japan) was adjusted to 10 mm. The post space and the sleeve were filled with the resin cement (i-TFC post resin, Sun Medical Co., Ltd., Shiga, Japan), and the post was inserted into the post space and simultaneously built up the core. The sleeve was then inserted into the post space as deeply as possible to ensure the best fit between the post and the sleeve. The specimen was subsequently cured with a light cure unit for 20 s. The abutment was manually prepared to a height of 5 mm from cervical region using a diamond bur (Figure 1B).

Group C (Sleeve composite group): the sleeve composites were prepared in accordance with the methods of a previous study [21], and used E-glass sheets (hollow) were adjusted to lengths of 10 mm. The MI post was cut to 15 mm for core restoration. Both the post space and the sleeve composite were filled with Gradia Core. The fitting of the sleeve composites and fiber posts and the abutment building were conducted in the same manner as in the i-TFC and MI groups, respectively (Figure 1C).

Group D (Custom-made tapered E-glass filling group): the post space was taken an impression with elastic impression material (Coltene/Whaledent, Altstätten, Switzerland) and the gypsum was cast in the model to form the root canal model. The E-glass sheet was dipped in BisGMA/TEGDMA, rolled up, fitted into the post space, and cured with light irradiation for 40 s to create the custom-made posts with an inner diameter of 1.5 mm. These were adjusted to lengths of 15 mm by shortening their coronal sections. A G-Premio bond was applied to the dentin in the root canal wall and the cervical shoulder regions at the top of the root for 10 s. The specimens were then cured with light irradiation for 20 s. Gradia Core was injected to fill the post space, and the post was adjusted into the space. The curing light was applied to the occlusal, buccal, and lingual sides for 20 s each. The abutment was manually prepared to a height of 5 mm from the cervical region using a diamond bur (Figure 1D).

Group E (Ferrule group): the remaining 10 teeth were adjusted to lengths of 17 mm, with 2-mm ferrules and a post space of 10 mm in depth, using a diamond bur with water coolant.

The lengths of the MI posts were adjusted to 15 mm. The posts were applied with Etchant Gel for 10 s, rinsed with water, and air-dried. The posts were then coated with Ceramic primer II, and a G-Premio bond was subsequently applied to the dentin in the root canal wall and the cervical shoulder regions at the top of the root for 10 s. Dual cure resin cement (Gradia Core) was injected to fill the post space, into which the post was then inserted and simultaneously built up the core. The specimens were cured with a light cure unit for 20 s. Following irradiation with the curing light on the occlusal, buccal, and lingual sides for 20 s each, the abutment was built up to a height of 5 mm using a diamond bur with water coolant (Figure 1E).

After core construction, the crowns of all specimens were made with direct composite resin restoration (G-ænial Anterior, GC Europe, Leuven, Belgium). The specimens were stored in the dark at 100% humidity and 37 °C for 24 h.

The teeth were then embedded into acrylic resin (Vertex self curing, Vertex-dental, Zeist, The Netherland). The polyvinyl chloride pipe was filled with acrylic resin, and a hole adjusted to the shape of the root was made. Specimens were embedded into the hole with BisGMA/TEGDMA and were cured with a light curing unit for 20 s. The specimens were embedded into the acrylic resin to a depth of 13 mm.

### 2.3. Loading Test

All specimens were fixed with the original gauge and were loaded at an angle of 45° to the long axis with a ball end (diameter: 2.0 mm) using the universal testing machine (Lloyd Lr3ok, Ametek, PA, USA). The loading test was conducted at a crosshead speed of 1.0 mm/min until fracture occurred (Figure 2).

The maximum load causing fracture was recorded for each specimen. Average fracture loads were compared with a one-way analysis of variance and Tukey’s honestly significant difference test (a = 0.05). Levene’s test was used to test the normality of data.

After the test, the modes of failure for all specimens, referred to here as the root fractures mode, were observed without microscopy. A two-examiner agreement was used to distinguish between favorable and unfavorable fractures. A favorable fracture is the type that ends above the cement-enamel junction (CEJ) (Figure 3I) and detachment of core construction (Figure 3II), whereas non-favorable fracture is the type that extends below the CEJ (Figure 3III). The fisher exact test with the Bonferroni correction was used to statistically present the favorable fracture (a = 0.05).

## 3. Results

The summary of the results with statistical differences is shown in Table 2. Group C exhibited significantly higher fracture strength than did the A and B groups (*p* < 0.05). There was no significant difference between the A, B, and D groups. Group E showed significantly higher fracture strength than did the other groups (*p* < 0.05).

Table 3 shows the results of fracture mode: 90% of the specimens in B, C, and E groups exhibited favorable fracture. This percentage dropped to 60% in D and 20% in A group. The proportions of favorable fractures were significantly high in all groups relative to group A (*p* < 0.05).

## 4. Discussion

The restoration of endodontically treated anterior teeth with flared root canals and with significant tooth structure loss poses a significant clinical challenge, and choosing the right post-core system restoration may be crucial to the treatment’s success. In our study, various restorative techniques were utilized to reinforce damaged endodontically treated anterior teeth. Our hypotheses were rejected because fracture behavior differed significantly between the restorative techniques used. Non-favorable tooth fracture was most frequently observed in group A, likely due to the relatively high amount of luting resin used in these specimens; specifically, the resin increased the polymerization shrinkage and compromised the adhesion [15,16]. Numerous studies have documented that high stresses can be imposed upon luting resins, particularly in the cervical area [22,23] and in vitro fatigue investigations have shown that post-luting resin microfracture or cracks is the initial failure mode that assists the development to catastrophic or non-favorable failure [24].

Various reports have demonstrated the usefulness of the i-TFC system, which can be attributed to the reduction in the luting resin layer and reinforcement effect of the sleeve [25,26,27]. Group B achieved almost the same loading value as that of group A; however, the former achieved a relatively higher percentage of favorable destructions because softer material and less luting resin was used in this group.

The new experimental sleeve composite, group C, had a higher loading value than the other groups as well as the highest percentage of favorable fractures. Although group C has a flexural modulus value close to that of group B, it has a higher maximum bending stress value than does group B [21]. Additionally, group C features multi-directional fibers [21], which increased Krechel’s factor and improved the reinforcement effect of this material [28]. Furthermore, as in the group B, the relatively less luting resin used in the new sleeve system contributed to reducing the rate of non-favorable fractures.

A decreased loading value and somewhat high non-favorable tooth fracture rate was observed in group D. The thickening of the sleeve composite layer caused stress concentration near the neck, resulting in an increase in flexural modulus. Furthermore, the adhesion was likely ineffective on account of this method being indirect.

Although the relatively diminished performance of the new sleeve method (group C) relative to group E indicates that improvement of methods is still warranted, our demonstration that the new sleeve system increases the loading value and the rate of favorable fracture indicates that this conservative method is clinically more effective than conventional methods for non-ferrule teeth. Our results are in agreement with previous studies, which showed that the presence of a ferrule is of high importance [29,30,31]. The purpose of the ferrule is to redistribute the occurring stress on the outer coronal third of the root, therefore possibly improving the fracture load and shifting the fracture pattern to a restorable one [31].

It important to highlight that in addition to the restorative techniques used, differences in loading values among the investigated groups might be attributed to other variables related to composite composition. There may be variations in filler loading, type and size, resin matrix composition, filler/fiber silanization quality, or the existence of some crack-resistance structures such as nano-clustering of fillers [32].

It is clearly shown in the literature that short-fiber reinforced composite (SFRC) when used as post-luting and core build-up resin leads to improved fracture behavior of endodontically treated teeth [23,32,33]. Therefore, it will be interesting to research how the new sleeve system and the SFRC work together to reinforce endodontically treated anterior teeth with flared root canals.

One of the limitations of our study is that static load-to-fracture testing was used without fatigue testing. Although static load-to-fracture tests mimic a sudden, greater force (like when trauma occurs), dynamic loading is more appropriate to study the mechanical consequences of the forces that act during ordinary chewing. An accelerated dynamic loading test represents a realistic compromise between the two extremes. Thus, fatigue testing should be carried out in the future in the same groups. In that case, the appropriate ferrule control groups to all groups need to be added.

The other limitation of this study is that the crown was made by direct restoration with resin composite. As its strength is lower than that achieved with the method of indirectly bonding metal, composite or ceramic crowns, future studies should consider the possibility of enhancing resistance to tooth fracture [34]. Further microleakage and adhesion testing studies are required to validate the finding of this laboratory study.

## 5. Conclusions

The new sleeve composite has moderate mechanical properties that significantly increase the fracture strength of non-ferrule teeth with flared root canals and decrease the rate of non-favorable fractures relative to other conventional methods. Our findings therefore suggest that the new sleeve composite could be clinically useful for the core construction of non-ferrule teeth.

## Figures and Tables

**Figure 1 polymers-14-04073-f001:**
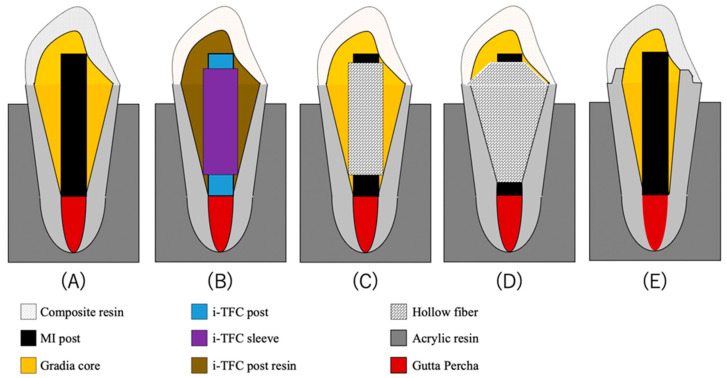
Long axial section of specimens with different restorative techniques (side view). (**A**) MI group, (**B**): i-TFC group, (**C**) Sleeve composite group, (**D**) Custom-made tapered E-glass filling group, (**E**) With ferrule group.

**Figure 2 polymers-14-04073-f002:**
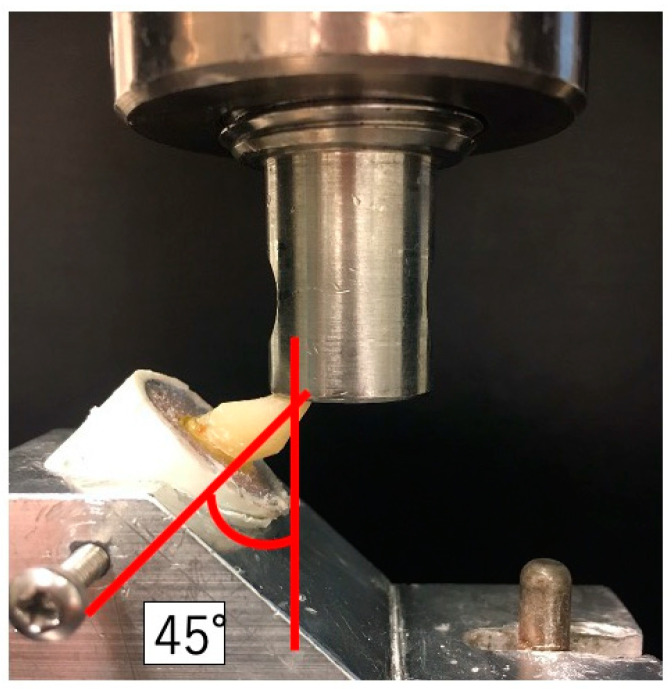
The photograph of the loading test set-up.

**Figure 3 polymers-14-04073-f003:**
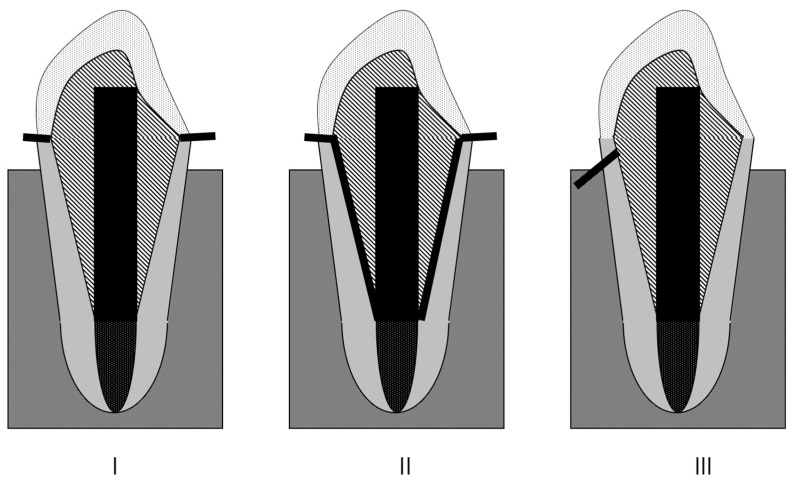
The types of fracture mode. (**I**) The type that fracture ends above the CEJ. (**II**) Detachment of core construction. (**III**) The type that extends below the CEJ.

**Table 1 polymers-14-04073-t001:** Restorative materials used during the study.

Materials	Manufacture	Lot Number	Characteristic	Composition
E-glass sheet	Hexcel	85M208171309		E-glass
i-TFC post	Sun Medical	SL1	Diameter: 1.5 mm	Glass fiber, Optical fiber UDMA-based matrix resin
i-TFC sleeve		SK1	Inner Diameter: 1.5 mm Outer Diameter: 2.0 mm	Glass fiber UDMA-based matrix resin
i-TFC post resin		SL12		Bis-MPEPP, PDMA, TEGDMA, UDMA, PI, Barium Silica Glass
i-TFC Bond		Bond; SR1R Brush; SR11		Bond: 4-META, Bis-MPEPP, PI, Acetone, Water, Silica Bond brushes: Sodyum p-toluenesulfinate, Aromatic amine
Gradia core	GC Europe	1804161		Fluoroaluminosilicate glass filler Bismethacrylate
MI post		20170728	Diameter: 1.5 mm	E-glass, PMMA
G-ænial Anterior		180605A		UDMA, Dimethacrylate co-monomer, slica
G-Premio Bond		1510291		10-MDP, 4-META, thiophosphate, methacrylate adicester, distilled water, acetone, photo initiators, silica fine powder
Ceramic primer II		1712062S		silane, MDP, ethanol
ScotchbondTM Universal Etchant	3M ESPE	564429		Water, Phosphoric Acid, Synthetic Amorphous Silica, Fumed, Crystalline free, Polyethylene Glycol, Aluminum Oxide

PMMA: Poly methyl methacrylate. UDMA: Urethane dimethacrylate. Bis-GMA: bisphenol A-glycidyl methacrylate. TEGDMA: Triethyleneglycoledimethacrylate. Bis-M EPP: 2,2′-bis(4-methacryloxy polyethoxyphenyl) propane. PDMA: Phenylene dimethacrylate. PI: Photo initiator. 10-MDP: 10-Methacryloyloxydecyl dihydrogen phosphate. 4-META: 4-Methacryloyloxy trimellitate anhydride.

**Table 2 polymers-14-04073-t002:** Fracture strength of the loading test.

Group	Fracture Load (N) MEAN ± SD
A (MI)	381.06 ± 85.0 a
B (i-TFC)	390.47 ± 112.32 a
C (Sleeve composite)	641.81 ± 105.54 b
D (Tapered E-glass filling)	443.55 ± 85.75 a
E (Ferrule)	843.79 ± 83.18 c

Different letter indicated significant different (*p* < 0.05).

**Table 3 polymers-14-04073-t003:** Distribution of fracture modes per group (*n* = 10).

Group	Favorable Fracture		Non-Favorable Fracture
	I	II	III
A (MI)	2	0	8
B (i-TFC)	8	1	1
C (Sleeve composite)	9	0	1
D (Tapered E-glass filling)	6	0	4
E (Ferrule)	9	0	1

## Data Availability

The data presented in this study are available on request from the corresponding author.

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
