# Peer review of "Influence of New Sleeve Composite on Fracture Behavior of Anterior Teeth with Flared Root Canals"

_polymers, 2022, doi:10.3390/polym14194073_

Round 1

Reviewer 1 Report

The presented study investigates the series of the endodontically treated teeth and their fracture strength and failure mode. Five groups treated with the different techniques were tested. The introduction is written cohesively. Testing materials, as well as methodology, are described in detail. The result part includes loading tests with the description. However, the evaluated study includes just one kind of experiment. Also, the discussion and conclusion parts are not comprehensive. The particular comments on each of the parts are listed below:

11.  The presented study described the loading test of investigated materials. It will be interesting to develop the research and compare the obtained results with more tests. It will provide new information, and the paper will be cohesive. Authors should try to perform the dynamic test - oscillatory strain measurement (ref. 15) or mechanical tests of the fillings like static tensile analysis, hardness tests, or bending module.

22..   Please add the description, of how the statistical significance was estimated.

33.   It is inconsequence regarding the name of the tested groups. In the “results” part are used A, B, C, D, and E (lines 170-171 and 175-176), and in the “discussion” part are used MI, i-TFC, Sleeve composition, and tapered E-glass filling (lines 193-211). It should be unified.

44.   It is not clearly described, why two kinds of fracture are favorable and one is non-favorable.

55.    It should be added the full description of the abbreviation after the first use (line 162, abbreviation CEJ)

66.   Experimental groups are described in detail, but the presented scheme of each specimen is not clear. Used fillings of the parts are similar, the legend is too small. It is hard to understand the picture (Figure 1). It is suggested to differentiate the parts of the fillings with different textures and colors.

77.   In the “discussion” part the authors suggest, that technical errors like an air bubble may influence the performed test (line 209). Is this difficulty appear just in case testing of the custom-made tapered E-glass filling group?

88.   The “discussion” part does not discuss the differences in chemical composition between each filling. It will be interesting if the components of individual composites influence the loading test. 

Reviewer 2 Report

The authors present an in vitro study evaluating sleeve composite's effect on anterior teeth' fracture behaviour. Four combinations of post and post space fillings were tested, plus one group with ferrule. The fracture behaviour was evaluated using a loading test, and the types of fractures were determined.

Table 1: I suggest removing the lines and lining up the information on the different columns to facilitate the readers’ comprehension.

How was the sample size determined?

The authors refer to “50 extracted endodontically treated anterior teeth”. What was the reason for extraction? In the title, the authors refer to flared root canals, which are not described in the materials and methods section. Also, since you re-instrumented the teeth, do you think that flared/not flared makes a difference in the obtained results?

All teeth presented only one canal? What type of teeth were included? Please add what teeth and how many of each type were included in each group.

What was the longest time the teeth were preserved in chloramine before being used?

Were the teeth evaluated for defects, such as fissures or fractures, before being used? If yes, how? Was this repeated after the canals were prepared?

How was the working length determined?

The dentin thickness was adjusted to 1mm in the cervical area on all walls?

Figure 1: please use capital letters (A, B, C, D and E) to match the text.

How was the data normality/not normality determined to decide the adequate statistical test to be used?

How were the fracture types evaluated? Did you use magnification?

None of the specimens presented a fracture between the post and the composite or between the sleeve and the composite?

Tables 2 and 3: I find the comparison between all groups inadequate. Group E is the control group of group A, but not to the other groups since the materials/techniques used are different. Comparing groups A and E, you can evaluate the importance of ferule to fracture resistance. Comparing groups A to D, you can determine the difference between materials and techniques.

Round 2

Reviewer 1 Report

I would like to recommend this paper for publication.

Author Response

Thank you very much for your review.

We are thankful for the time and energy you expended.

Reviewer 2 Report

The authors present an in vitro study evaluating sleeve composite's effect on anterior teeth' fracture behaviour. Four combinations of post and post space fillings were tested, plus one group with ferrule. The fracture behaviour was evaluated using a loading test, and the types of fractures were determined. The performed modifications improved the manuscript quality, but further corrections are needed.

Table 1: please indicate the meaning of the diameter symbol in the table caption.

Please add the information to the manuscript on how the sample size was determined.

Line 72: how was this evaluated (the one canal)? Using x-ray? Please add this information to the manuscript.

Lines 86-87: did you use magnification to access fissures and fractures? Please add this information to the manuscript.

Figure 1: please correct “gutta percha”.

Lines 162-163: please add the information to the manuscript that fracture modes were evaluated without magnification.

Lines 166-168: please clarify the sentence.

I suggest adding to the study limitations the lack of appropriate ferrule controls for all groups, which could support further comparisons between the groups.
